# Syngeneic N1-S1 Orthotopic Hepatocellular Carcinoma in Sprague Dawley Rat for the Development of Interventional Oncology-Based Immunotherapy: Survival Assay and Tumor Immune Microenvironment

**DOI:** 10.3390/cancers15030913

**Published:** 2023-01-31

**Authors:** Bongseo Choi, Jason Pe, Bo Yu, Dong-Hyun Kim

**Affiliations:** 1Department of Radiology, Feinberg School of Medicine, Northwestern University, Chicago, IL 60611, USA; 2Department of Biomedical Engineering, McCormick School of Engineering, Northwestern University, Evanston, IL 60208, USA; 3Robert H. Lurie Comprehensive Cancer Center, Northwestern University, Chicago, IL 60611, USA; 4Department of Biomedical Engineering, University of Illinois at Chicago, Chicago, IL 60607, USA

**Keywords:** hepatocellular carcinoma (HCC), rodent HCC animal model, cancer immunotherapy, interventional oncology

## Abstract

**Simple Summary:**

Interventional oncology (IO) approaches have been highly effective in treating hepatocellular carcinoma (HCC) locoregionally. Recently revealed anti-cancer immunity of IO-based immunotherapies proposed the promise of enhanced immunotherapy. However, the development of those therapies is still in its infancy with limited preclinical animal models. Preclinical animal models in IO-based immunotherapy research are the key resources to navigate the optimal procedure and efficacy of newly proposed IO-based immunotherapy approaches.

**Abstract:**

Rodent HCC rat models provide advantages for interventional oncology (IO) based immunotherapy research compared to other established larger animal models or mice models. Rapid and predictable tumor growth and affordable costs permit the formation of a compelling preclinical model investigating novel IO catheter-directed therapies and local ablation therapies. Among orthotopic HCC models, the N1-S1 orthotopic HCC model has been involved in many research cases. Suboptimal tumor induction rates and potential spontaneous regression during tumor implantation procedures discouraged the use of the N1-S1 HCC model in IO-based immunotherapies. Here, N1-S1 HCC models were generated with a subcapsular implantation of two different number of N1-S1 cells using a mini-laporatomy. Tumor growth assay and immunological profiles which can preclinically evaluate the therapeutic efficacy of IO-based immunotherapy, were characterized. Finally, an N1-S1 HCC rat model generated with the proposed procedure demonstrated a representative immune suppressive HCC tumor environment without self-tumor regression. The optimized syngeneic N1-S1 HCC rat models represent an essential tool for pre-clinical evaluation of new IO immunotherapies for the treatment of HCC.

## 1. Introduction

Interventional oncology (IO) local therapies are practicing minimally invasive treatment of hepatocellular carcinoma (HCC) [1,2,3,4,5]. A total of 50–60% of HCC patients might receive IO image-guided local therapies [6]. Recent studies reveal that IO local therapies treating the primary HCC might induce the shrinkage of untreated distant tumors, this is known as an abscopal effect [7,8]. Finding various IO local therapy mediated immune modulation and anti-cancer immunity is now in great interest for the potential combination with immune checkpoint inhibitor (ICI) immunotherapy [9,10]. Various combinations of immunogenic IO local therapy and ICI immunotherapy are ongoing in the clinical trials, to improve the overall therapeutic outcomes and survival benefit versus monotherapy of HCC [11,12,13]. Recent studies are showing that the response to ICI immunotherapy significantly relies on a pre-existing immune suppressive tumor microenvironment (TME) [14]. Unfortunately, HCC establishes unique immune suppressive TME that is considered as the main challenge to achieve a satisfactory level of therapeutic efficacy. Dynamic cancer cellular interaction, cytokine release, and signal cascades build a strong immune suppressive TME to be dysfunctional adaptive and innate anti-cancer immune responses in HCC. Various preclinical studies to understand immunosuppressive mechanisms and evaluate the efficacy of combinational immunotherapies that can convert immune suppressive TME are critical to further advance the promising combination of IO and immunotherapy of HCC. The development of preclinical animal models will be the key resource to understand those immune reactions and tumor progress, as well as anti-cancer immune mechanism of innovate combination IO and immunotherapy modalities for the HCC. Large animals including rabbit, pig, and woodchuck models have been used for testing IO hepatic artery procedures and monitoring tumor size changes. Although they can provide the anatomical scale required to develop IO-based treatments, these models are costly and require a significant amount of training and support staffs. More importantly, there is a lack of commercial antibodies that can characterize immune responses for immunotherapy research. Although mice HCC models can be useful to characterize immuno-biological responses with a well-established antibody portfolio, the inherently small anatomies of mice are the main limitation in testing IO therapeutic research. Rat HCC models can be a feasible option for the preclinical combination IO and immunotherapy research [15,16,17]. Among rat HCC models, DEN-induced HCC, McA-RH7777 cells (Morris [18] hepatoma model), and N1-S1 cells (Novikoff [19] hepatoma models) have been most frequently employed to generate HCC rat models. A diethylnitrosamine (DEN)-induced Wistar rat autochthonous HCC model [20] demonstrated recapitulation of the hepatocyte injury cirrhosis malignancy evolutionary cycle, as is seen with human HCC with tumor hypervascularity. However, the main limitation of this model is the difficulty of tumor response analysis with uncontrolled metastasis and a required long period for proper tumor establishment (approximately 3 months for the DEN rat model vs. 7–14 days for orthotopic rat models). The Morris hepatoma model was generated in Buffalo rats following exposures to N-2-fluorenylphthalamic acid. The generated tumor was used to develop the McA-RH7777 cell line. A syngeneic HCC can be initiated with orthotopic injection of McA-RH7777 cells in Buffalo rats. The model is also compatible with Sprague Dawley rats [21]. Advantages of this model include a high inoculation rate and well-developed tumoral vascular structures. A shortcoming is the lack of Buffalo rat availability, currently it is not commercially available [22]. The Novikoff hepatoma model originated from exposure of Sprague Dawley (SD) rats to 4-dimethylaminoazobenzene yielding the N1-S1 cell line. N1-S1 tumors are produced through orthotopic hepatic injection of syngeneic N1-S1 cells. Well-established syngeneic N1-S1 HCC rat models can be valuable to characterize various immune responses of preclinical combination of IO and immunotherapy strategies [23]. However, the controversial results of tumor growth curve and potential for spontaneous regression in various cell implantation procedures impeded its application as a tumor model in IO [24]. To evaluate if the N1-S1 HCC rat model can be used for testing IO-based immunotherapy applications, this study investigated a consistent HCC tumor growth after the subcapsular N1-S1 implantation. N1-S1 cell amount dependent tumor generation and volume changes were analyzed for 21 days post-N1-S1 implantation. The following TME immune status of N1-S1 rat model was characterized with flowcytometry analysis using antibodies that can be used for detecting rat immune cells. The immune baseline of N1-S1 rats is discussed with regard to clinical HCC immune status in the tumor microenvironment.

## 2. Materials and Methods

### 2.1. Tumor Cell Line

The N1-S1 hepatoma cell line was acquired from American Type Culture Collection (CRL-1604, Manassas, VA, USA), which were kept at 37 °C in a humidified 5% CO_2_ atmosphere, and cultured in Dulbecco’s Modified Eagle Media (Gibco, Grand Island, NY, USA) supplemented with 10% fetal bovine serum (Gibco, Grand Island, NY, USA) and 1% penicillin/streptomycin (Gibco, Grand Island, NY, USA). Cells were subcultured every three days. Cell viability was determined by 0.4% trypan blue staining (Invitrogen, Carlsbad, CA, USA) and machine counting with a CountessTM Automated Cell Counter (Invitrogen, Carlsbad, CA, USA), which confirmed greater than 90% N1-S1 cell viability prior to tumor implantation.

### 2.2. N1-S1 Cell Implantation in Sprague Dawley Rats

All procedures involving experimental animals were performed in accordance with protocols approved by the Institutional Animal Care and Use Committee of Northwestern University. Sprague Dawley (SD) rats (Charles River Laboratories, Wilmington, MA, USA) weighing between 250 and 330 g (4–6 weeks old) were used for this study. All animals had ad libitum access to a water standard laboratory diet. For subcapsular N1-S1 cell implantation, SD rats were anesthetized with 2% isofluorane with 3 L/min oxygen, animals were monitored for depth of anesthesia via respiratory rate and foot pinch retraction. Once adequately anesthetized, preoperative buprenorphine was administered at 0.05 mg/kg subcutaneously into the nape of the neck. The animals were then shaved in the abdominal region and clorhexidine scrub solution (VWR, Radnor, PA, USA) was then applied to the shaved region with a sterile gauze pad. Using the inferior xiphoid process as landmark, a scalpel was used to make a 1 cm midline abdominal incision inferiorly from the subxiphoid process with great care not to injure the underlying viscera. Once the incision was completed, one sterile cotton-tipped applicator was gently inserted into the incision site to carefully isolate the liver parenchyma from the surrounding intraabdominal viscera. Using the two cotton-tipped applicators, the liver was gently lifted anteriorly through the incision site, exposing the left lateral lobe. Next, either 2.5 × 10^6^ (2.5 million) or 5 × 10^6^ (5 million) freshly harvested N1-S1 cells diluted in sterile 1x Dulbecco’s Phosphate Buffered Saline (Mediatech, Corning, Manassas, VA, USA) with a total volume of 0.1 mL was very slowly injected 5 mm into the injection site, over the course of at least 30 s, immediately deep to the hepatic capsule. Upon injection, a white bulge appeared on the liver surface. A 0.5 × 0.5 cm piece of BloodStop hemostatic gauze matrix (LifeScience PLUS, Mountain View, CA, USA) was then placed at the injection site and pressed with light pressure for at least 10 s, before the inoculation needle was slowly retracted, while continuing to keep steady pressure at the injection site for another 30 s. This slow retraction method and constant pressure was performed to prevent extravasation of N1-S1 cells outside of the liver parenchyma. The exposed liver was gently placed back into the abdominal cavity. Incision closure was made in two layers using 4-0 absorbable Vicryl sutures (Ethicon, Somerville, NJ, USA). Meloxicam at 2 mg/kg was then administered subcutaneously in the nape of the neck immediately post-op, and at day 1 post-op. The animals were allowed to recover in cages with food and water available ad libitum. Once the animals were determined to be in stable condition, they were returned to the animal facility. Tumors were allowed to grow while the animals were observed daily for any signs of distress.

### 2.3. Monitoring Tumor Growth

Each week for 3 weeks, the rats were anesthetized with 2% isofluorane and 3 L/min oxygen and monitored via abdominal ultrasound. Mindray M7 ultrasound machine (Mindray North America, Mahwah, NJ, USA) with a Mindray L14-6a Linear Ultrasound Transducer (Mindray North America, Mahwah, NJ, USA). Ultrasound jelly was placed liberally on the rats’ abdomens and gently imaged using the ultrasound transducer. At 3-week post-implantation, both the 2.5 million and 5 million cell groups were also imaged with 1.5 T Aera MRI System (Siemens, Munich, Germany) and small 4-channel flex coil (TIM Systems, Siemens Medical Solutions, Erlangen, Germany). T2-weighted images were collected for monitoring the size of HCC. MR scans were performed in both coronal and axial orientations using a gradient-echo sequence. Tumor volume (V) was calculated as V = a^2^b/2 (where a is width, b is length, and a ≤ b). For comparison purposes, the tumor volume was normalized by its initial volume as V/V0 (V0 was the volume of the tumor at the time that catheterization). Prior to MRI scans, the animals were anesthetized with ketamine (Ketaset, Fort Dodge Animal Health, Fort Dodge, IA, USA) at 80 mg/kg and xylazine (Isothesia, Abbot Laboratories, Chicago, IL, USA) at 4 mg/kg via intraperitoneal injection.

### 2.4. Flow Cytometry Analysis

Rat spleens and HCC tumors were harvested after euthanization. Spleens and HCC cells were homogenized separately and stained with the following antibodies: CD3-PE, CD4-PEcy7, CD8-FITC, CD-25-APCcy7, FoxP3-PECF594, PD-1, and PD-L1 (Becton Dickinson, Franklin Lakes, NJ, USA). BD LSRFortessa 6-Laser flow cytometer (Becton Dickinson, Franklin Lakes, NJ, USA). Data analysis was conducted with Flowjo software (Flowjo LLC, Becton Dickinson, Ashland, OR, USA). Gating strategies, a list of antibodies/fluorophores, and additional information are provided in Appendix A).

### 2.5. Histology Analysis

Harvested tissues including tumor and normal hepatocytes in a proper lobe underwent fixation with 10% formalin (ThermoFisher Scientific, Waltham, MA, USA) for 2 weeks followed by paraffin embedding, 10-micron tissue sectioning, and hematoxylin and eosin staining, Ki-67, TUNEL, CD34, and PD-L1 with the assistance of Northwestern University Mouse Histology and Phenotyping Laboratory (Chicago, IL, USA). [15] Microscopy slides were visualized with light microscopy (Olympus CK Inverted Phase Tissue Culture Microscope (Shinjuku, Tokyo, Japan) and Amscope image capture software (United Scope LLC, Irvine, CA, USA)). TUNEL positive cells in each group were quantified by counting 500 cells in a random area [25].

### 2.6. Statistical Analysis

Data are expressed as mean ± standard deviation (SD) or standard error (SE). Differences between the values were assessed by Student’s *t*-test. A one-way ANOVA (Kruskal–Wallis test) was used to compare tumor size and progression.

## 3. Results and Discussion

### 3.1. N1-S1 Cell Implantation and HCC growth in Sprague Dawley Rats

Cultured N1-S1 cells were successfully implanted in the hepatic capsule of left-lobes with a mini-laparotomy procedure (Figure 1a). All procedures of 24 rats were completed without any complications (bleeding, infection, and abnormal activities) or failures (mortality) by researchers with more than two years of experience. The tumor induction rates of 2.5 × 10^6^ N1-S1 and 5 × 10^6^ N1-S1 cells were both 100 % (each group 12/12 and 12/12). Our results demonstrate that the direct implantation of 2.5 or 5 million N1-S1 cells in the liver generated reproducibly N1-S1 HCC tumors in 3-week post-implantation. No spontaneous tumor regression was observed in the period. As shown in Figure 1b, the tumors appeared as hypoechoic nodules in all the rats from 7 days post-implantation. Tumors continued to grow during the 3 weeks after implantation and remained hypoechoic. The tumor growth curve from US imaging showed an exponential growth of tumors in both 2.5 and 5 million N1-S1 cells-implanted rats (Figure 1c). Consistent tumor growth of 5 × 10^6^ N1-S1 cell-implanted rats reached to 533 mm^3^ of the mean tumor volume. When 2.5 × 10^6^ N1-S1 cells were implanted, the mean tumor size at 21-day posttreatment decreased to 218 mm^3^. No tumor regression was observed in both groups for 21 days of post-implantation. The final measured tumor volume with US imaging at 21 days post-treatment was also consistent with the tumor size measured in MRI T2 weighted images and gross tumor tissue images (Figure 1d,e). H&E-stained slides of the tumors grown with both 2.5 and 5 million N1-S1 cells showed a clear tumor region in the liver (Figure 1f). Tumor rims between HCC and normal parenchymal hepatocytes were clearly shown.

### 3.2. Histological Analysis of N1-S1 HCC Rats

All 24 of the 24 rats that were inoculated demonstrated histopathological evidence of tumor growth (Figure 2a,b). H&E staining of liver tissue samples revealed solid masses of hyperplastic, irregular hepatocyte growth, atypical, multi-nucleated parenchymal cells which grew beyond the basement membrane. Tumor cell proliferation and developed vasculatures confirmed successful N1-S1 tumor development. H&E-stained slides of the tumors grown with both 2.5 and 5 million N1-S1 cells showed anaplastic round cells with necrotic region at the tumor cores (Figure 2a). Ki67 histology demonstrated aggressively proliferating tumor cells in both groups of 2.5 and 5 million N1-S1 implanted rats. CD34 positive signals showing tumoral micro-vessels were well-distributed in the tumors of both 2.5 and 5 million N1-S1 cells-implanted groups. TUNEL staining showed a strong positive signal in necrotic tumor regions and no significant apoptosis signal in non-necrotic tumor area. Peripheral hepatic regions primarily showed hepatocytes without significant metastatic tumor cells (Figure 2b). Ki67, CD34 and TUNEL staining of the peripheral hepatic tissues showed no significant difference between 2.5 and 5 million N1-S1 cells implanted groups.

### 3.3. Immunological Analysis of N1-S1 HCC Rats

Generated N1-S1 HCC rats could not recapitulate all immunological components of clinical HCC TME. However, the base line of the immune response was useful for evaluating the immune reaction after various IO immunotherapies. For the preclinical immunological studies to characterize main effector T cells of immunotherapies, a list of antibodies (Appendix A) was used for the flowcytometry analysis. Here, CD8+, CD4+, and Treg were representatively characterized with flowcytometry analysis (Figure 3). Considering clinical interest on PD-1/PD-L1 immunotherapy, PD-L1 expression of N1-S1 HCC was also measured. On flow cytometry, CD8+ cytotoxic T lymphocyte (CTL) accumulation in tumor after 3 weeks of implantation was approximately 28–30% in tumor and 17–26% in spleen, respectively (Figure 3a). These data insist that CD8+ CTLs were accumulated into the HCC during the N1-S1 tumor growth. The CD4+ T cell composition were also at a high level in both groups (Figure 3b). However, Tregs (Foxp3) from CD4+ T cells were accumulated into the HCC implanted with N1-S1 cell implantation at 3 weeks post-implantation (Figure 3c). The 5 × 10^6^ N1-S1-implanted group showed significantly higher Treg (%) than the 2.5 × 10^6^ N1-S1 cells-implanted group. The lower ratio of CTLs and Treg of 5 × 10^6^ N1-S1-implanted group might indicate significantly decreased anti-cancer immunity toward the HCC tumor, resulting in an increased tumor growth rate. The significantly high PD-L1 expression level on both 2.5 and 5 million N1-S1-implanted groups (Figure 3d) also supports the theory that HCC tumor growth escaped from anti-cancer immunity of rats. Functionally activated T lymphocytes including CTLs expressed immune checkpoint PD-1 [26,27], which is the corresponding receptor of PD-L1 on cancer cells. To control the excessive immune responses, immune-suppressing cells including regulatory T cells suppress CTLs by PD-L1 overexpression [28]. Generated N1-S1 tumors also expressed an intensive amount of PD-L1 on their surface, which enabled the active immune suppression of CTLs in TME (Figure 3d and Appendix A). This result supports that HCC SD rat models with N1-S1 might be utilized to test the immune checkpoint inhibitor-related IO immunotherapies.

### 3.4. Tumor Microenvironment and HCC Tumor Response to aPD-L1 Immunotherapy in N1-S1 HCC Rat Model

A maximum clinical dose of aPD-L1 (10 mg/kg) [29] was intravenously (IV) administered to the N1-S1 HCC rats (Figure 4a). After 14 days of the systemic aPD-L1 ICI immunotherapy treatment, the tumor growth and immune changes were characterized and compared to non-treated control N1-S1 HCC rats. The tumor size changes after the IV injection of aPD-L1 were not significantly different with the non-treated control group (Figure 4b). Only a small number of CD3 positive T cell population were infiltrated in the tumor treated with IV injection of aPD-L1 as similar level with the non-treated control group (Figure 4c). High amounts of immune-suppressive cells including MDSC (21.5%) and Tregs (37.4%) of non-treated HCC were not effectively changed after IV injection of aPD-L1 (Figure 4c). Histology images and quantitative analysis of tumor sections stained with H&E and TUNEL also showed no significant difference of cancer cell death between the groups of IV injection of aPD-L1 and control, as shown in the tumor progression for 2 weeks posttreatment (Figure 4d). The maximum dosage of aPD-L1 for the N1-S1 rats was used with a consideration of strong immune-suppressive TME of HCC. However, the excessive aPD-L1 administration could not modulate the immune-suppressive TME representing the high amount of MDSC and Tregs of N1-S1 HCC for suppressing tumor growth, as shown in the clinical results [30].

## 4. Discussion

Preclinical animal models are the key resource to navigate the optimal procedure and efficacy of newly proposed combinational IO and immunotherapy approaches. Due to the high complexity and sensitivity of immune system, and limited amount of clinical immune response data for the treatment of HCC, preclinical immunological studies are valuable to set up important factors for the clinical trial design and approaches on the development of IO-based immunotherapy. Although mice models are generally considered as the best preclinical animal models to investigate in-depth details of immunological events, IO research utilizing various catheters, local ablation probes, and imaging are not applicable in the small anatomy of mice. Rodent HCC rat models provide more advantages compared to other established larger animal models or mice models. Rapid and predictable tumor growth and affordable costs permit the formation of a compelling preclinical model investigating novel IO catheter directed therapies and local ablation therapies [31,32,33]. For pre-clinical evaluation of new IO immunotherapies, syngeneic HCC rat models, which have a functional immune system, represent an essential tool for the research. Among various HCC rat models, N1-S1 HCC in SD rats could be a reliable preclinical syngeneic HCC animal model for IO immunotherapy studies. However, the low tumor induction rate and potential spontaneous tumor regression after an ultrasound image guided percutaneous implantation of N1-S1 cells hindered its active uses as a preclinical in vivo animal model. As demonstrated with the consistent growth curve of the tumor volume in both 2.5 × 10^6^ and 5 × 10^6^ N1-S1 cell groups, the N1-S1 rat models in our study can be used as a syngeneic rat model of liver cancer to monitor tumor growth and anticancer immune research for 2–3 weeks. The 5 × 10^6^ N1-S1 group was considered a more aggressive tumor growth that could be used for the analysis in less than 2 weeks.

Recent advances in the understanding of tumor biology and immune checkpoint molecules (PD-1/PD-L1) have provided novel therapeutic strategies using immune checkpoint inhibitors (ICI). ICI-mediated immunotherapy has emerged as an effective and promising treatment for HCC [34,35]. However, recent phase I/II trials of ICI immunotherapy for HCC patients showed only a modest response rate of 19% [35,36]. ICB with nivolumab and pembrolizumab did not show significant benefit in randomized phase III trials in HCC [37,38]. The inferiority of aPD-L1 immunotherapy in HCC compared to other solid tumors is mainly attributed to tumor resistance or ignorance with the immune suppressive TME. The development of HCC promotes the immune checkpoints overexpression, infiltration of immune-suppressive regulatory T cells (Tregs), and myeloid-derived suppressor cells (MDSCs) in the TME [39,40,41]. Thus, the current systemic ICI immunotherapy injecting anti-PD1 or anti-PD-L1 antibodies may not be effective to activate anti-cancer immunity of HCC nor to elicit durable clinical benefit in HCC [37,38,42,43]. Although pre-clinical HCC rat models are frequently used in IO research, immune-suppressive TME of HCC rat models has not been reported well. In our study, highly immune-suppressive TME and the clinical challenge of systemically delivered ICI immunotherapy in HCC was observed in the N1-S1 HCC rat model generated with 2.5 × 10^6^ N1-S1 cell implantation. Those immune suppressive TME with systemic aPD-L1 therapy demonstrates the current challenge of aPD-L1 immunotherapy for the treatment of HCC. The immune responses in N1-S1 HCC rats were similar with the clinical situation. Although this is a limited study with one maximum clinical dosage of aPD-L1, the N1-S1 HCC rat model demonstration of immune suppressive TME and immune responses after systemic aPD-L1 immunotherapy provides a feasibility of testing IO-based immunotherapeutic strategy and immunological information for investigating HCC tumor responses in the preclinical stage. Specifically, it is expected to be used for screening and development of various IO-based immunotherapy along with finding prognostic markers and imaging markers for the clinical IO-based immunotherapies of HCC. Recent progress of IO-based immunotherapy of HCC is demanding various information of fundamental mechanism, toxicity, and immunological tumor microenvironment to translate promising IO-based immunotherapies that have been demonstrated in an early stage. In the multiple clinical translational process, the N1-S1 HCC model will be a useful preclinical animal platform to find crucial information such as the dosage, sequence, and timing of combinational IO-based immunotherapy.

## 5. Conclusions

Immune suppressive TME in HCC is considered to be the main challenge to achieve a satisfactory level of immuno-therapeutic efficacy. Considerable therapeutic potential of various combinations of immunogenic IO local therapy and ICI immunotherapy has been identified to improve the overall therapeutic efficacy of HCC. Various preclinical studies to evaluate the potential therapeutic efficacy and to understand immune response of combinations of IO therapy and immunotherapy are critical to further advance the promising combinational IO and immunotherapy of HCC. Preclinical animal models will be essential surrogate platforms to evaluate the immune reaction and tumor responses as well as anti-cancer immune mechanisms of newly proposed IO and immunotherapy combinational modalities for the HCC. However, syngeneic animal models and information about immune profiles of those animal models are very limited. The N1-S1 orthotopic HCC model is a well-known tumor model, and thus it must be involved in various IO research cases. However, their uses are limited by suboptimal tumor induction rates, and the potential for spontaneous regression. Here, our demonstrated subcapsular tumor implantation method with over 2.5 million N1-S1 cells reproducibly generated N1-S1 HCC tumor during a period of 3 weeks without the host innate immune system mediated self-tumor regression. The syngeneic N1-S1 HCC rat model also provided immune profiles of N1-S1 rats for the preclinical IO-based immunotherapy research. Immune status of N1-S1 tumors by the implantation of 2.5 and 5 million N1-S1 cells consistently showed high activity of T-regulatory (Treg) cells and low amounts of infiltrating CD4+ and CD8+ T lymphocytes. Monitored PD-L1 levels of N1-S1 HCC were also a measurable level required to test the efficacy of PD-1/PD-L1 ICI immunotherapies. Finally, when we tested systemic aPD-L1 ICI immunotherapy in the N1-S1 HCC rats, the characteristic immune suppressive MDSC and Treg upregulation of HCC was observed, i.e., a representative immunosuppressive TME similar with the findings in clinical HCC. Our demonstrated protocols about N1-S1 cell implantation, subsequent consistent tumor growth, and the immune profiles of N1-S1 rat models comparable with the immune-suppressive characteristics of clinical HCC will provide an important resource to conduct preclinical IO and immune-oncology research.

## Figures and Tables

**Figure 1 cancers-15-00913-f001:**
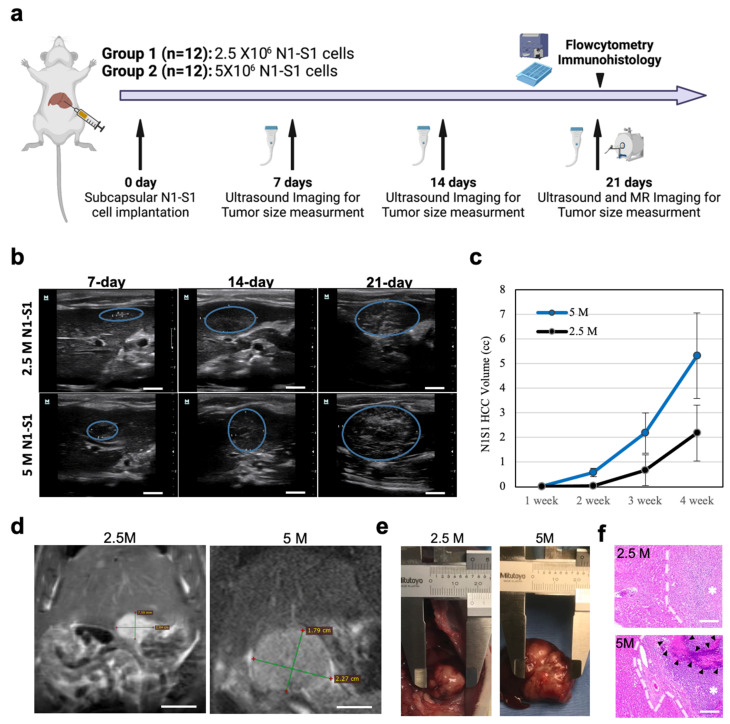
(**a**) Schematic summary of experimental groups and procedure. Rats in Group 1 were implanted with 2.5 × 10^6^ N1-S1 cells and rats in Group 2 were implanted with 5 × 10^6^ N1-S1 cells. The tumor size was tracked with ultrasound and MR imaging for 3 weeks. Harvested tumor and liver parenchyma tissues were analyzed with immunohistochemistry. (**b**) Ultrasound imaging and tumor size tracking of N1-S1 rats generated with 2.5 and 5 million N1-S1 cell implantations. Blue circles are tumors. Scale bars are 1 cm. (**c**) Time-dependent tumor volume changes measured in ultrasound imaging. (**d**) MR T2 weighted images of N1-S1 tumors generated by 2.5 or 5 × 10^6^ N1-S1 cells at 3 weeks post-implantation. Scale bars are 1 cm. (**e**) Gross tissues images from each experimental group. (**f**) H&E-stained tissues showing the tumor borders (white dotted lines) from each group. (*: tumor region and black arrows: tumor central necrosis region). Scale bars are 100 μm.

**Figure 2 cancers-15-00913-f002:**
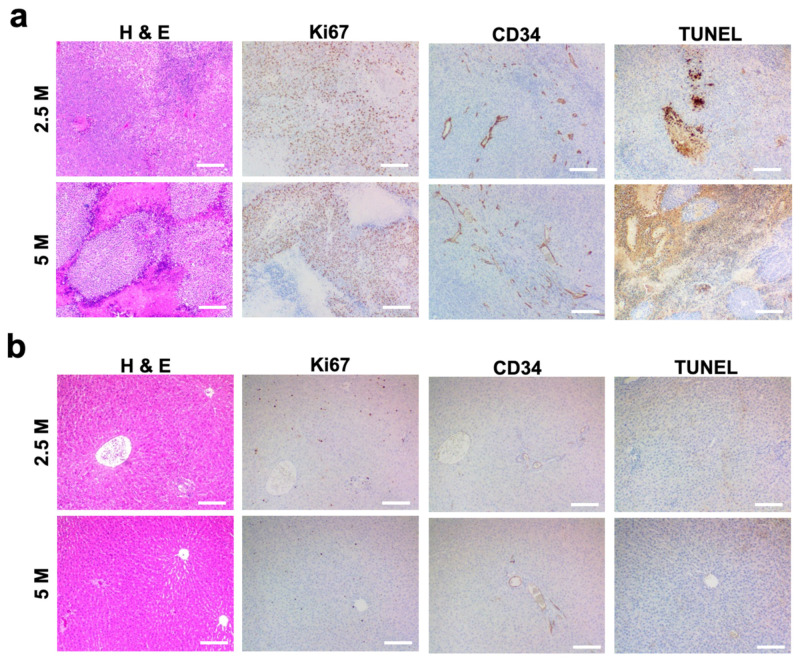
Histological analysis of HCC tumor generated with a subcapsular N1-S1 cells (2.5 and 5 million N1-S1 cells) implantation in Sprague Dawley rats. H&E histology and immunohistology (Ki67, CD34, and TUNEL) images of (**a**) tumor and (**b**) peripheral hepatic region. Scale bars are 150 μm.

**Figure 3 cancers-15-00913-f003:**
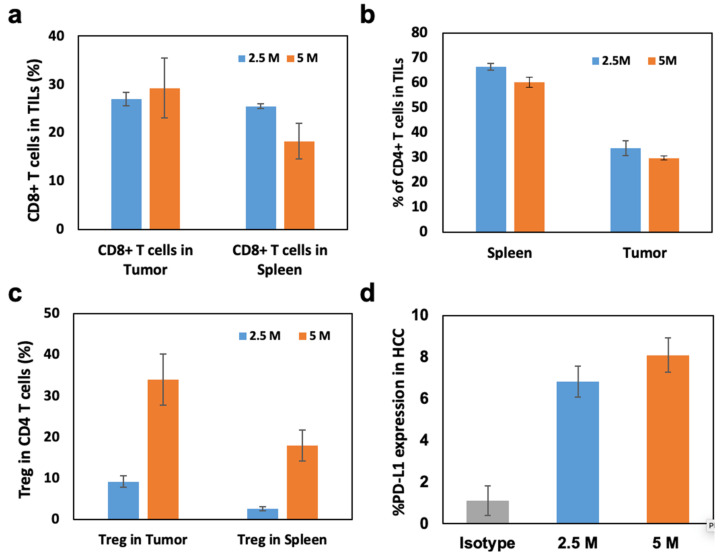
(**a**) CD8+ T cells, (**b**) CD4+ T cells, and (**c**) regulatory T cells (Tregs) in tumor infiltrated lymphocytes (TILs) of N1-S1 HCC tumors with 2.5 and 5 million N1-S1 cell implantations. (**d**) PD-L1 expression levels of N1-S1 HCC tumors generated with 2.5 or 5 million N1-S1 cells-implanted rats and isotype rats.

**Figure 4 cancers-15-00913-f004:**
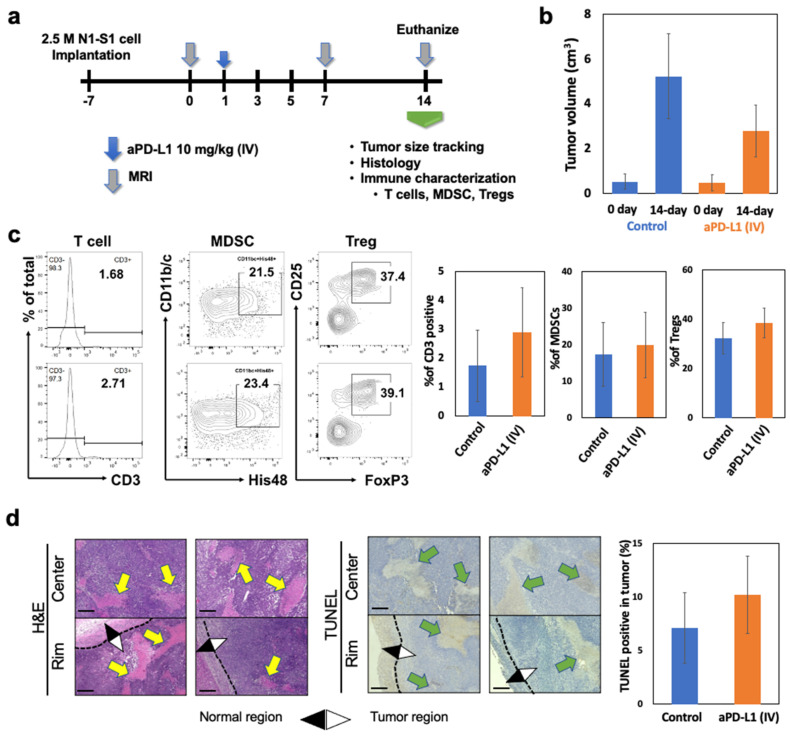
Immune suppressive TME of HCC hindered the therapeutic response of aPD-L1 immunotherapy. (**a**) Schedule of immunotherapy. After tumor diameter diagnosed by MRI imaging (bigger than 1 cm), 10 mg/kg of aPD-L1 was injected through systemic (IV). Tumor size changes after 14 days of systemic immunotherapy. (**b**) Tumor size changes after 14 days of systemic (IV) immunotherapy. (**c**) Flowcytometry analysis of T cells (CD3+), MDSCs (CD11b/c+His48+) and Tregs (CD3+CD4+CD25+Foxp3+) from TILs of HCC tumor after IV injection of aPD-L1 (10 mg/kg) or equivalent amount of PBS. (**d**) H&E staining, TUNEL staining of HCC tumor slices after aPD-L1 immunotherapy. Yellow arrows in H&E panel indicates the none-nucleus necrosis and green arrow in TUNEL panel indicates the TUNEL positive stained area. Scale bars are 150 um. Data was obtained from three independent samples.

## Data Availability

Data is available from the corresponding author upon reasonable request.

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
