# Peer review of "Syngeneic N1-S1 Orthotopic Hepatocellular Carcinoma in Sprague Dawley Rat for the Development of Interventional Oncology-Based Immunotherapy: Survival Assay and Tumor Immune Microenvironment"

_cancers, 2023, doi:10.3390/cancers15030913_

Round 1

Reviewer 1 Report

The authors performed flow cytometry analysis and histological analysis of tumor-infiltrating lymphocytes and other components of tumor immune microenvironment in an advanced rat model of hepatocellular carcinoma. The study design is sound; however, the manuscript requires some improvements to become more beneficial to the readers, namely:

-Introduction: Are there any large animal models (e.g. pig) used in this field of study? If so, compare the pros and cons.

-I am missing a clearly formulated testable scientific hypothesis at the end of Introduction. This biological question should be then clearly answered in Conclusions.

-Material and Methods: The surgical part is described well and in a reproducible manner. However, I am missing the list of antibodies used for immunohistochemistry, including dilution, pretreatment etc. Also the Table S1 is missing an important imformation - the antibodies were used for identification of what cell phenotypes? Do not let your readers guess. Also, the histological methodology is not described sufficiently. I am missing the following: Tissue sample and organ weight and dimensions, tissue block dimensions, multilevel sampling - number of tissue blocks processed, samples sectioned, microscopic fields of view observed and evaluated, numbers of cells counted?

-Results and Discussion: It is not a good idea to start this section with discussing and evaluatin (lines 194-206) prior to presenting the results. It has to be perfeclty clear, what are your findings and data and what is the interpretation.

-All micrographs should have scale bars with dimensions stated in the figure legends.

-Avoid repeating methodology in the Results section, e.g. lines 258-260.

-Provide your primary data as e-Supplements along with the manuscript.

-Discussion: Make a separate paragraph summarizing the translational potential of your model into human medicine. There are papers analyzing T- and B-cells in the invasive borders of resected human hepatocellular caricinoma - how does your animal model compare to the human HCC?

Reviewer 2 Report

HCC is increasingly followed both in our country and in the world, ranking second among all cancers and fifth among cancer-related deaths. Difficulties in diagnosis, follow-up and treatment led physicians to different searches. In cases where TARE and TAKE applications in the recent periods are insufficient, systemic treatments are included in the implementation. Especially in smart drug models, great benefits are obtained. In recent years, immunotherapy agents have been developed in many cancer models, especially HCC. The syngeneic rat model is thought to be the most suitable model for these studies. It is believed that with this beautifully designed and studied model, great advances will be made in the treatment of HCC and answers will be found to many unmet questions and problems. I would like to thank you, our esteemed authors, who organized and carried out this beautiful work, and wish you continued success. I trust that your work will benefit both students, assistants and professors a lot. It will illuminate the way for further steps to be taken, rather than greater success in the treatment of HCC.

Reviewer 3 Report

The authors generated Rat HCC model with subcapsular implantation of N1S1 cells using a mini laparotomy, and pathohistological and immunologically analyzed HCC tumor. Unfortunately, although it is said "Interventional Oncology-based Immunotherapy", "Interventional" techniques such as catheterization have not been established.  They should rewrite the title and the introduction section. The results section has a repeat of the same content as the intro section. This is unnecessary.

minor: The cell line name is sometimes described as N1-S1, and sometimes as N1S1. The names should be unified. In the method section, the age of the rats should be written. HCC histology should also be presented with a high-magnification microscopic image. A "hepatocellular carcinoma" model cannot be established without showing viable hepatoma cells.

Reviewer 4 Report

The present study developed an optimized syngeneic N1-S1 hepatocellular carcinoma (HCC) rat model for preclinical evaluation of new interventional oncology (IO) immunotherapies for the treatment of HCC. The model demonstrated a representative immune suppressive HCC tumor environment without self-tumor regression. The study is of great importance and clinical potential. Some minor points are listed as below.

1. The results are suggested to be divided into several subtitles.

2. The discussion section should be independently presented.

3. The conclusions section should be simplified and highlight the novel findings.

Round 2

Reviewer 1 Report

The authors addressed the points raised during the first round of the review.  The present form of the manuscript is more informative, better structured and documented and more readable. 

Reviewer 3 Report

Adequate revision, no more comments. Thank you.